# Clinical Question Influence on Radiation Dose of Cardiac CT Scan in Children

**DOI:** 10.3390/children9081172

**Published:** 2022-08-05

**Authors:** Theodor Adla, Martin Kočí, Vojtěch Suchánek, Zuzana Šalagovičová, Michal Polovinčák, Lukáš Mikšík, Jan Janoušek, Miloslav Roček

**Affiliations:** 1Department of Radiodiagnostic and Interventional Radiology, Institute for Clinical and Experimental Medicine, 140 21 Prague, Czech Republic; 2Department of Radiology, University Hospital Motol and 2nd Faculty of Medicine, Charles University, 150 06 Prague, Czech Republic; 3Children’s Heart Centre, University Hospital Motol and 2nd Faculty of Medicine, Charles University, 150 06 Prague, Czech Republic

**Keywords:** paediatric cardiac CT, radiation dose, dose-length product, clinical question

## Abstract

Background: To assess the impact of different clinical questions on radiation doses acquired during cardiac computed tomography in children. Methods: A total of 116 children who underwent cardiac CT on a third-generation dual-source CT scanner were included. The clinical questions were divided into three main categories: the extent of scanning in the *z*-axis, coronary artery assessment and cardiac function assessment. Radiation dose values represented as a dose-length product (DLP) in mGy*cm were recorded from the CT scanner protocols. Results: There were significantly higher doses in cases with cardiac function assessment (median DLP 348 versus 59 mGy*cm, *p* < 0.01) and in cases with coronary artery assessment (median DLP 133 versus 71 mGy*cm, *p* < 0.01). Conclusion: The most important factor was the assessment of cardiac function, where the median radiation dose was 4.3× higher in patients with a request for cardiac function assessment. We strongly recommend that clinical requests for cardiac CT should be carefully considered in the paediatric population.

## 1. Introduction

ECG-synchronized cardiac CT due to high spatial and temporal resolution is ideal for non-invasive assessment of the heart. Due to technological advances and radiation dose reduction protocols, the overall radiation dose has decreased in recent years [1]. However, radiation burden from this examination is not negligible, especially in children due to higher radiosensitivity [2,3].

There are several factors that affect radiation dose from a CT examination, including the type of scanning mode (retrospective or prospective ECG synchronisation), lowering of the electrical voltage of the X-ray tube and also the craniocaudal range of scanning [1,4,5,6]. Nevertheless, selecting the appropriate scanning protocol to minimize radiation dose is not only the decision of the radiologist or CT technician, but also depends, in many cases, on the clinical question. Ultimately, lowering the radiation dose itself should not outweigh the clinical benefit of the examination. Clinical questions for paediatric cardiac CT vary and may encompass anatomical evaluation of the entire chest, coronary artery evaluation and the assessment of cardiac function with measurement of the volumes and ejection fractions of the left and right ventricle, and eventually combinations of some or all of these tasks.

There are multiple studies dealing with cardiac CT and radiation dose [7,8,9,10,11,12,13]. However, these studies address the issues of radiation dose and technical and diagnostic quality, but none of them compare radiation dose with clinical questions. This study analyses the influence of different clinical questions on the radiation dose of paediatric cardiac CT.

## 2. Materials and Methods

This is a retrospective, single-centre study performed at the University Hospital Motol in Prague. We included all children and adolescents below the age of 19.

In total, 116 children (41 females, 75 males) were examined (Table 1). The main diagnosis was congenital heart disease (74%). Other diagnoses were less common: coronary anomaly (6%), ischemic heart disease (5%), connective tissue disease (5%), infection including infective endocarditis (4%), vasculitis (3%), two cases with arrhythmia and one case each for trauma, thrombosis and tumour. Grouping of patients according to intervals based on body weight according to the European Guidelines on Diagnostic Reference Levels for Paediatric Imaging was omitted in this article [14]. The main reasons for this simplification were a relatively small number of examinations and an effort to keep the article clear and concise.

A third-generation dual-source CT scanner (Somatom Force, Siemens Healthineers, Forchheim, Germany) was used for all examinations. Periodical quality control tests were performed on a regular basis in the range recommended by the vendor, approved by the local clinical medical physicist and recommended by laws and regulations in the Czech Republic, including a long-term stability test and a daily operational stability test. The scanning modes used were: prospective ECG-triggered high-pitch helical scanning (61×), retrospective ECG-gated helical scanning with current modulation (49×), retrospective ECG-gated helical scanning without current modulation (4×), and prospective ECG-triggered axial scanning (2×). Selection of the scanning mode was based on the clinical question and on the supervising physician’s decision. The voltage was set as low as recommended for the given weight (70–90 kV); however, in selected cases of known implanted metallic stents or pulmonary Melody valves, the voltage was set higher, at 100–150 kV. The current was adjusted by CT scanner software using topogram-based automatic tube current selection (CareDose4D, Siemens Healthineers).

### 2.1. Clinical Questions

The principal diagnosis of every patient was noted. However, more important than the principal diagnosis itself are the clinical questions influencing the selection of the CT protocol and scanning mode. There are many different clinical questions, and, in many cases, there was more than one question. However, in view of CT protocol selection, they could be simplified to three main categories of clinical questions:Extent of scanning in the *z*-axis (heart only): Yes or no. This differentiates between examinations that cover only the heart (from the tracheal carina to the caudal edge of the heart) and examinations that include the aortic arch or the entire chest. A longer extent of scanning leads to a higher radiation dose.Coronary artery evaluation: Yes or no. This differentiates between examinations where one of the questions is about anatomy, stenosis or compression of the coronary arteries versus examinations without the need for precise evaluation of the coronary arteries. The relevant assessment of the coronary arteries depends on the quality of the images, especially the absence of motion artefacts. Prospective single-shot high-pitch mode requires a lower heart rate (below <60 bpm), which is sometimes impossible in children. Therefore, we assume that retrospective gating mode is more often selected for coronary CT angiography, and this could lead to a higher radiation dose.Cardiac function assessment: Yes or no. This differentiates between examinations in which one of the clinical questions is about the measurement of cardiac ventricles and the calculation of the ejection fraction of the right and/or left ventricles. Assessment of function requires scanning through the whole heart cycle using retrospective gating mode, which therefore leads to a higher radiation dose than in examinations at a particular cardiac cycle phase only.

### 2.2. Radiation Dose

Radiation dose values were recorded from the CT scanner protocol stored in the hospital picture archiving and communication system (PACS). As recommended by the European Guidelines on Diagnostic Reference Levels for Paediatric Imaging, a standard cylindrical CT phantom with a diameter of 32 cm was used for calibration [14]. The dose-length product (DLP) in mGy*cm was used for the representation of a radiation dose because DLP is a good representation of the total amount of ionising radiation applied during the examination. Size-Specific Dose Estimate (SSDE) values are not available in the patient dose protocol of the used CT scanner [15].

Other quantities are used to express the radiation dose in medical imaging with ionising radiation including the effective dose [15,16,17]. Furthermore, there are different conversion factors for the calculation of the effective dose from the DLP for different age groups, and moreover, these factors differ among different studies and recommendations [3,13,18,19,20]. Therefore, because of the age and weight heterogeneity of our group of patients, we did not calculate the effective dose from the DLP.

### 2.3. Statistical Analysis

Statistical analysis was performed with MedCalc Statistical Software version 19.4.0 (MedCalc Software Ltd., Ostend, Belgium; https://www.medcalc.org, accessed on 22 December 2020). Mann–Whitney test for independent samples was used for DLP and Student’s *t*-test was used for other variables.

## 3. Results

The total DLP presented as a median (minimum, maximum) was 101 (6, 1751) mGy*cm.

### 3.1. Extent of Scanning in the Z-Axis

There was no significant influence of the extent of scanning on the resulting radiation dose. The DLP of the examinations limited to the heart was not significantly higher—about 1.54× in comparison to examinations of the entire chest (median DLP 129 versus 84 mGy*cm) (Table 2 and Figure 1).

### 3.2. Coronary Artery Evaluation

The clinical question dealing with the evaluation of the coronary arteries had a statistically significant influence on the resulting radiation dose. The DLP of coronary artery examination was significantly higher (1.87×) compared to examinations without it (median DLP 133 versus 71 mGy*cm) (Table 3 and Figure 2).

### 3.3. Cardiac Function Assessment

The clinical question about cardiac function assessment has a statistically significant influence on the resulting radiation dose (median DLP 348 versus 59 mGy*cm); however, there was also a significant difference between age, height, weight, BSA and BMI within the group of patients with and without cardiac function assessment (Table 4).

There was no significant difference in the evaluated parameters, except for radiation dose (median DLP 84 versus 358 mGy*cm), after the elimination of selection bias using an age limit above 8.1 y.o. The question on cardiac function assessment increases the radiation dose of cardiac CT examination by 4.3× (Table 5 and Figure 3).

## 4. Discussion

The clinical question about coronary arteries and cardiac function assessment caused a statistically significant increase in the radiation dose acquired from cardiac CT in paediatric patients. The most important factor was cardiac function assessment, where the median radiation dose was 4.3× higher in patients with cardiac function assessment requests than without. In the case of coronary artery assessment, there was 1.87× higher median radiation dose than without.

To the best of our knowledge, there is no similar study that compares radiation dose with the clinical question. However, an explanation of our results can be derived from previous studies. For example, Podberesky et al. compared retrospective and prospective ECG-gated cardiac CT scanning using anthropomorphic phantoms and found that scanning in the retrospective mode increases the radiation dose by 3.8× [7]. Cardiac function assessment requires retrospective mode in all cases, but coronary artery or entire chest examination can be performed in both modes.

Comparison of the total DLP, scanners and scanning protocols with other studies is summarized in Table 6. The total DLP is similar to studies where they used a combination of retrospective and prospective scanning modes. Only Mainel et al. mentioned selection of retrospective ECG-gated scanning based on myocardial function assessment and showed a similar range of total DLP. Several studies with a third-generation dual-source CT scanner (128-slice DCST) showed that the prospective high-pitch scanning protocol provides scans with very low radiation doses, with sufficient examination quality for evaluation of the heart and vessels in congenital heart disease [8,11,12,15,17]. However, this mode can be used in cases where cardiac function assessment was not necessary. Alternatively, a step and shoot prospective ECG synchronized protocol could be used in the case of CT machines without a high-pitch mode. This mode also provides low radiation doses with sufficient image quality [13].

### Study Limitations

Due to a limited number of examinations and in an effort to keep the structure of presented data clear and concise, we did not divide the patients into groups as per the European Guidelines on Diagnostic Reference Levels for Paediatric Imaging. This limitation is obvious in the case of cardiac function evaluation, where there was cardiac CT examination selection bias due to easier evaluation of cardiac function by echocardiography in younger and smaller children. To eliminate this selection bias, we had to state an age limit in order to compare similar groups. This limit was derived from the mean minus SD of the group of patients with cardiac function assessment (13.3–5.2) who were 8.1 years or older. After this elimination of selection bias, there was no significant difference in the evaluated parameters except radiation dose, as mentioned above (Table 5 and Figure 3).

## 5. Conclusions

The factor increasing radiation the most was the cardiac assessment, where the median radiation dose was 4.3× higher in patients with cardiac function assessment than without. Based on these results, we strongly recommend that clinical requests for cardiac CT should be carefully considered. Radiation-free methods, such as echocardiography and magnetic resonance imaging, should be used for cardiac function assessment whenever possible.

## Figures and Tables

**Figure 1 children-09-01172-f001:**
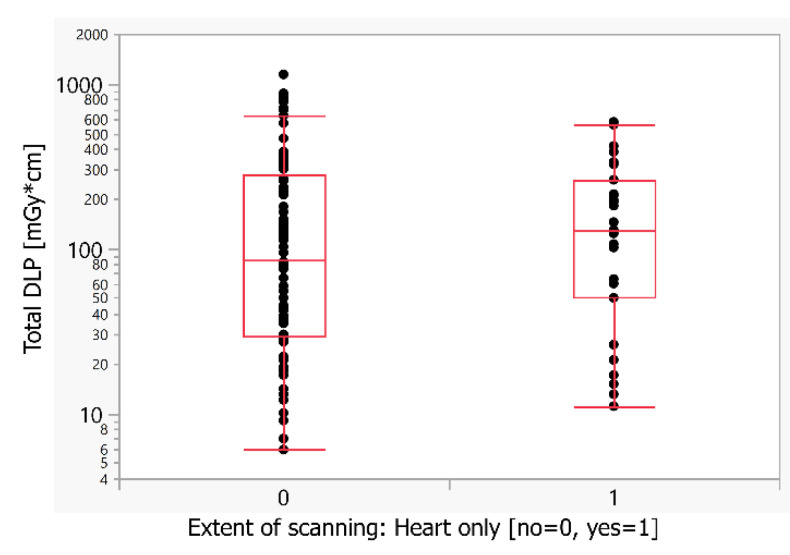
Comparison of the radiation dose for the extent of scanning in the *z*-axis. DLP—dose length product (mGy*cm).

**Figure 2 children-09-01172-f002:**
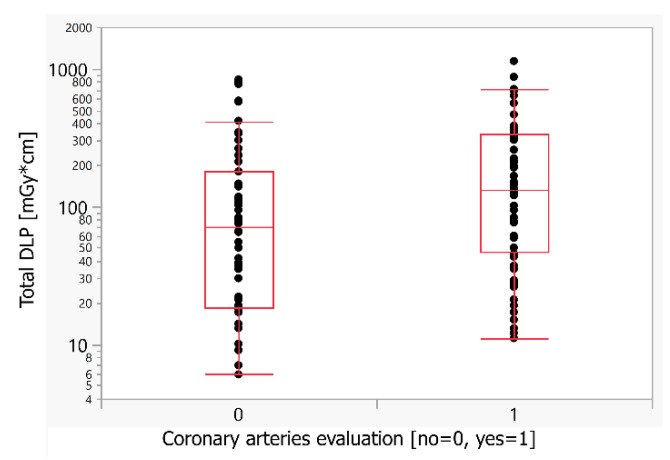
Comparison of the radiation dose for the coronary artery evaluation. DLP—dose length product (mGy*cm).

**Figure 3 children-09-01172-f003:**
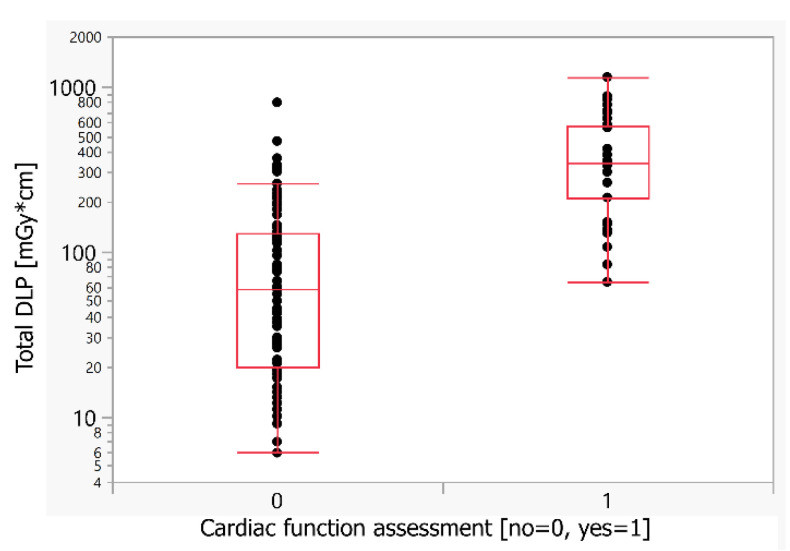
Comparison of the radiation dose for the cardiac function assessment, limited to patients aged > 8.1 y.o. to exclude selection bias due to patient age. DLP—dose-length product (mGy*cm).

**Table 1 children-09-01172-t001:** Patients’ characteristics (116 examinations).

Characteristic	Mean ± SD (Min–Max)
Age	10.6 ± 5.9 years (50 days–18.9 years)
Height	138.2 ± 38.0 cm (54–200)
Weight	40.1 ± 24.8 kg (3.7–103)
BSA	1.22 ± 0.55 (0.24–2.30)
BMI	18.3 ± 4.3 (11.4–33.3)

**Table 2 children-09-01172-t002:** Influence of the extent of scanning in the *z*-axis on the radiation dose of cardiac CT scan in children.

Parameter	Heart Only: No (n = 89)	Heart Only: Yes (n = 27)	Comparison	*p* Value
Age [years]	10.3 ± 5.9	11.5 ± 5.7	1.2 CI (−1.3 to 3.8)	0.3322
Height [cm]	136.1 ± 38.9	145.1 ± 34.4	9.0 CI (−6.6 to 24.7)	0.2520
Weight [kg]	39.2 ± 25.5	43.3 ± 22.9	4.1 CI (−6.3 to 14.5)	0.4272
BSA [m^2^]	1.19 ± 0.56	1.30 ± 0.51	0.11 CI (−0.12 to 0.34)	0.3257
BMI [kg/m^2^]	18.2 ± 4.6	18.6 ± 3.1	0.4 CI (−1.1 to 1.9)	0.6131
DLP [mGy*cm]	84 (6, 1133)	129 (11, 584)	1.54×	0.3957

The DLP value is presented as a median (minimum, maximum), and the comparison value is presented as a ratio. The other variables are presented as mean and standard deviation with comparison as a difference, with a 95% confidence interval. For *p* values, the Mann–Whitney test for independent samples is used for DLP and the Student *t*-test is used for the other variables. BSA—body surface area; BMI—body mass index; DLP—dose-length product; CI—confidence interval.

**Table 3 children-09-01172-t003:** Influence of coronary artery evaluation on the radiation dose of cardiac CT scan in children.

Parameter	Coronary Arteries: No (n = 56)	Coronary Arteries: Yes (n = 60)	Comparison	*p* Value
Age [years]	10.1 ± 6.6	11.0 ± 5.1	0.9 (−1.3 to 3.1)	0.4010
Height [cm]	131.8 ± 42.7	144.2 ± 32.2	12.3 (−1.7 to 26.3)	0.0833
Weight [kg]	37.7 ± 26.5	42.4 ± 23.2	4.8 (−4.4 to 13.9)	0.3069
BSA [m^2^]	1.15 ± 0.60	1.28 ± 0.49	0.14 (−0.07 to 0.34)	0.1873
BMI [kg/m^2^]	18.0 ± 4.1	18.5 ± 4.5	0.6 (−1.0 to 2.2)	0.4731
DLP [mGy*cm]	71 (6, 831)	133 (11, 1133)	1.87×	0.0072

The DLP value is presented as a median (minimum, maximum), and the comparison value is presented as a ratio. The other variables are presented as mean and standard deviation with comparison as a difference, with a 95% confidence interval. For *p* values, the Mann–Whitney test for independent samples is used for DLP and the Student *t*-test is used for the other variables. BSA—body surface area; BMI—body mass index; DLP—dose-length product; CI—confidence interval.

**Table 4 children-09-01172-t004:** Influence of cardiac function assessment on radiation dose of cardiac CT scan in children. All patients.

Parameter	Function: No (n = 85)	Function: Yes (n = 31)	Comparison	*p* Value
Age [years]	9.5 ± 5.8	13.3 ± 5.2	3.8 (1.5 to 6.1)	0.0013
Height [cm]	132.4 ± 39.5	154.0 ± 28.3	21.6 (8.31 to 34.8)	0.0018
Weight [kg]	36.0 ± 24.0	51.5 ± 23.8	15.6 (5.5 to 25.6)	0.0030
BSA [m^2^]	1.13 ± 0.55	1.46 ± 0.47	0.34 (0.13 to 0.55)	0.0017
BMI [kg/m^2^]	17.6 ± 3.9	20.2 ± 4.9	2.7 (0.7 to 4.6)	0.0089
DLP [mGy*cm]	59 (6, 796)	348 (65, 1133)	5.9×	<0.0001

The DLP value is presented as a median (minimum, maximum), and the comparison value is presented as a ratio. The other variables are presented as mean and standard deviation with comparison as a difference, with a 95% confidence interval. For *p* values, the Mann–Whitney test for independent samples is used for DLP and the Student *t*-test is used for the other variables. BSA—body surface area; BMI—body mass index; DLP—dose length product; CI—confidence interval.

**Table 5 children-09-01172-t005:** Influence of cardiac function assessment on radiation dose of cardiac CT scan in children. Selection limited to patients aged > 8.1 y.o. for exclusion of bias.

Parameter	Function: No (n = 48)	Function: Yes (n = 27)	Comparison	*p* Value
Age [years]	13.8 ± 3.4	14.9 ± 3.6	1.0 (−0.6 to 2.6)	0.2177
Height [cm]	160.4 ± 21.7	162.9 ± 16.0	2.4 (−7.0 to 11.9)	0.6070
Weight [kg]	51.5 ± 19.3	57.1 ± 20.1	5.6 (−3.6 to 14.8)	0.2313
BSA [m^2^]	1.50 ± 0.38	1.59 ± 0.35	0.09 (−0.08 to 0.27)	0.3065
BMI [kg/m^2^]	19.2 ± 3.8	20.9 ± 4.9	1.5 (−0.8 to 3.7)	0.1695
DLP [mGy*cm]	84 (11, 796)	358 (83, 1133)	4.3×	<0.0001

The DLP value is presented as a median (minimum, maximum), and the comparison value is presented as a ratio. The other variables are presented as mean and standard deviation with comparison as a difference, with a 95% confidence interval. For *p* values, the Mann–Whitney test for independent samples is used for DLP and the Student *t*-test is used for the other variables. BSA—body surface area; BMI—body mass index; DLP—dose-length product; CI—confidence interval.

**Table 6 children-09-01172-t006:** Comparison of total DLP, scanners and scanning protocols with other studies.

Study	Total DLP (mGy*cm)	Scanner	Scanning Mode
Adla (this study)	101 (6, 1751)	128-slice DSCT	Prospective ECG-triggered high-pitch helical, retrospective ECG-gated helical with current modulation, retrospective ECG-gated helical without current modulation or prospective ECG-triggered axial. Selection of the scanning mode based on clinical question and supervising physician’s decision.
Barrera 2019 [12]	98.29 ± 66.02 (17.6, 204.9) **	128-slice DSCT	Prospective ECG-triggered high-pitch helical
Hou 2017 [13]	15.29 ± 1.91 * (Lower dose) 20.11 ± 2.13 * (Standard dose)	64-slice MDCT	Prospective ECG-triggered axial
Liu 2016 [10]	19.71 ± 10.5 *	64-slice DSCT	Prospective ECG-triggered axial
Koplay 2016 [11]	15.6 ± 9.6 *	128-slice DSCT	Prospective ECG-triggered high-pitch helical
Rompel 2016 [19]	5.33 ± 3.05 * (128-slice DSCT) 9.17 ± 4.05 * (64-slice DSCT)	64-slice DSCT 128-slice DSCT	Prospective ECG-triggered high-pitch helical
Habib Geryes 2016 [20]	Mean varies from 28.4 to 189.2 (3 different protocols and 4 age groups)	64-slice MDCT	Retrospective ECG-gated, prospective ECG-triggered
Meinel 2015 [9]	67 (1, 1788)	64-slice MDCT 64-slice DSCT	Retrospective ECG-gated, prospective ECG-triggered, or an adaptive sequential technique depending on the patients’ heart rate and rhythm and whether or not an evaluation of myocardial function was indicated
Ghoshhajra 2014 [8]	107.0 (44.5, 282.3)	64-slice MDCT 64-slice DSCT 128-slice DSCT	Retrospective ECG-gated helical, prospective ECG-triggered axial, or prospective ECG-triggered high-pitch helical. ECG-gating was selected per the scanner’s availability, scan indication and per the supervising physician’s discretion.

The DLP is presented as a median (minimum, maximum), * mean ± standard deviation, or ** mean ± standard deviation (minimum–maximum). ECG—electrocardiogram, MDCT—multi-slice CT, DSCT—dual-source CT.

## Data Availability

Data will be made available by the corresponding author upon reasonable request.

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
