# Peer review of "Clinical Question Influence on Radiation Dose of Cardiac CT Scan in Children"

_children, 2022, doi:10.3390/children9081172_

Round 1
Reviewer 1 Report
Well written, concise enough. Clear description of the scientific background, study design and research objectives.
The results and conclusions are reported in a clear and exhaustive manner, underlining the major innovations of the submitted study.
Innovative work because it evaluates the dose absorbed during a cardio CT exam on the basis of the clinical question, suggesting practical advice in daily diagnostic activity.
Author Response
Dear Reviewer,
Thank you very much for your kind review of our article. We really appreciate the time you spent on this.
Sincerely
Theodor Adla et al.
Reviewer 2 Report
General Comments
The aims of this work were “Analyses the influence of different clinical questions on the radiation dose of paediatric cardiac CT”. I think that this kind of paper is important to children readers, however, I have a series of questions and suggestions throughout the text. So, after responding favourably to all these questions and suggestions, in my opinion, the paper can be accepted for publication.
Some specific comments:
Materials and Methods:
1. Page 2. Line 53. You should expand the description about system used. It must also indicate whether the X-ray system is subject to a quality assurance program and what quality control tests are performed.
2. Page 2. Line 54. Classification is very generic for classify the radiation dose in patients. Should have used to EC 2018 recommendations. This is one major limitation of the study. More explanation and justification are needed here. Reference: EC 2018 European Commission. European guidelines on diagnostic reference levels for paediatric imaging.Radiation protection No 185. Publications Office of the European Union.
3. Page 3. Line 29. In different reports of the International Commission Radiological Protection (ICRP) and International Commission on Radiation Units and Measurements (ICRU), indicate the dosimetric quantities and radiological protection quantities to be use. You must develop more because you are using DLP and effective dose and not CTDIvol?.
You must submit the bibliographic references for quantities used. Some reference to consult: ICRP 2007 The Recommendations of the International Commission on Radiological Protection ICRP Publication 103 Ann. ICRP 37 1-332 ICRP 2017 Diagnostic Reference Levels in Medical Imaging ICRP Publication 135 Ann. ICRP 46 1–143 ICRU 2005 Patient dosimetry for x rays used in medical imaging ICRU Report 74 J. (Bethesda, MD: International Commission on Radiological Units and Measurements)
Results: 1. Re-write this section. In this section just leave the text that presents your table or figure. Transfer the other commented texts to the discussion section.
Discussion: 1. Re-write this section. You must necessarily comment or compare your results (all tables and figures) inside or with other articles. It is suggested to add a section describing all the limitations of the study
Author Response
Dear Reviewer,
Thank you very much for your effort to improve the article. We really appreciate the time you spent on this.
Please see the attachment for our response
Sincerely
Theodor Adla et al.

Round 2
